# Contrastive Learning Based on Transformer for Hyperspectral Image Classification

**Xiang Hu** [1][iD]**, Teng Li** [2,3]**, Tong Zhou** [1,2]**, Yu Liu** [3] **and Yuanxi Peng** [1,*]

[1] The State Key Laboratory of High-Performance Computing, College of Computer, National University of Defense Technology, Changsha 410073, China; huxiang@nudt.edu.cn (X.H.); zhoutong09@nudt.edu.cn (T.Z.)

[2] Beijing Institute for Advanced Study, National University of Defense Technology, Beijing 100020, China; liteng09@nudt.edu.cn

[3] College of Advanced Interdisciplinary Studies, National University of Defense Technology, Changsha 410073, China; liuyu11@nudt.edu.cn

[*] Correspondence: pyx@nudt.edu.cn

**Abstract:** Recently, deep learning has achieved breakthroughs in hyperspectral image (HSI) classification. Deep-learning-based classifiers require a large number of labeled samples for training to provide excellent performance. However, the availability of labeled data is limited due to the significant human resources and time costs of labeling hyperspectral data. Unsupervised learning for hyperspectral image classification has thus received increasing attention. In this paper, we propose a novel unsupervised framework based on a contrastive learning method and a transformer model for hyperspectral image classification. The experimental results prove that our model can efficiently extract hyperspectral image features in unsupervised situations.

**Keywords:** deep learning; transformer; unsupervised hyperspectral image classification; contrastive learning

## 1. Introduction

Compared with general images, hyperspectral images can provide more abundant pixel-level spectral information, since they contain hundreds of spectral bands. This enables the pixel-level classification of hyperspectral images, which then led to hyperspectral image classification becoming a hot topic in remote sensing. Nowadays, it is widely used in many fields, such as crop estimation [1], soil salinity estimation [2], and mineral mapping [3].

Recently, deep learning has performed remarkably in computer vision tasks, such as image classification [4,5], facial recognition [6], and objection detection [7]. Inspired by this, many deep-learning-based methods for hyperspectral image classification have been proposed, such as deep recurrent neural networks (RNNs) [8], three-dimensional convolutional neural networks (3DCNNs) [9], the deep feature fusion network (DFFN) [10], and spectral-spatial residual network (SSRN) [11]. With enough labeled training samples, many deep-learning-based methods can provide high classification accuracy.

Similar to computer vision, most of the state-of-the-art deep-learning-based methods are based on CNN architectures because CNN has achieved the best performance in these two areas. However, unlike computer vision, the state-of-the-art CNN for hyperspectral image classification is based on a 3D convolutional architecture instead of a 2D convolutional architecture. The 3D CNNs require much more computational resources, and researchers have spent much time designing efficient 3D CNN models for hyperspectral image classification. Additionally, CNNs fail to capture the sequence attributes of spectral signatures well, particularly middle- and long-term dependencies.

The redundant spectral information of hyperspectral images leads to high costs of hyperspectral data acquisition. Labeling a large number of hyperspectral data is unrealistic, requiring extensive human resources. Unsupervised learning methods can work without

labels, so this challenge can be solved. Thus, research on unsupervised hyperspectral image classification is urgently required.

Unsupervised learning aims to extract information from data without labels. According to the mainstream view, unsupervised methods can be divided into representative learning and discriminative learning. Most of the representative learning methods are based on two models: an autoencoder [12] and generative adversarial network (GAN) [13]. These two models both aim to map the training data to a certain distribution mode. According to their distribution, we may obtain fake samples that are similar to real data or useful feature extractors. For hyperspectral image classification, the stacked sparse autoencoder (SSAE) extracts sparse spectral features and multiscale spatial features using the autoencoder [14]. Additionally, GAN [15] was used for hyperspectral image classification. Furthermore, the 3D convolutional autoencoder (3DCAE) [16], GAN-assisted CapsNet (TripleGAN) [17], and many other representative-learning-based models have been proposed for hyperspectral image classification.

Except for representative learning, discriminative learning is a method for collecting information from unlabeled data. Contrastive learning is a popular discriminative learning method. Unlike representative learning, contrastive learning aims to discriminate different data instead of obtaining the data distribution feature. It requires much less computational resources than the representative learning method. After being trained by comparing different samples, the discriminator can be applied to downstream tasks, such as image recognition [18] and object detection [19].

Contrastive learning has achieved great success in the computer vision field. However, some challenges remain when applying contrastive learning to hyperspectral image classification. Firstly, the data augmentation methods commonly used in the computer vision field are not applicable to the hyperspectral image classification field. For example, color distortion is a typical data augmentation method used for general images. However, when used for hyperspectral image classification, color distortion disrupts the spectral information in hyperspectral images. Therefore, this method is unsuitable for hyperspectral image augmentation. Secondly, the models used in contrastive learning, mostly 2D CNNs, for computer vision tasks are not applicable in hyperspectral image processing. CNNs are not able to mine the sequence attributes of spectral signatures well.

To introduce contrastive learning to hyperspectral image classification, we handled these two problems in this study. Firstly, we devised a useful data augmentation method for hyperspectral image classification. Secondly, since 3D CNNs require much more computational resources than 2D CNNs, we used a transformer model instead of convolutional neural networks as the feature extractor. The transformer architecture is well-designed for effectively processing and analyzing sequential data. We only adjusted the model parameters, which had little impact on the model computation.

In this article, we introduce bootstrap your own latent (BYOL), a state-of-the-art contrastive learning framework, to hyperspectral image classification using a transformer architecture. After training under the unsupervised condition, the transformer model can extract features from the input hyperspectral image. We use a SVM model to obtain excellent classification performance with a small percentage of features and corresponding labels for training. The contributions of our work are as follows:

1. We introduce contrastive learning to extract hyperspectral image features in the unsupervised situation, which removes the manual labeling costs. As many contrastive methods rely on large amounts of negative samples to work well, which greatly reduces the training efficiency, we use BYOL as the contrastive learning framework. BYOL avoids the need for negative examples. Moreover, we adjust the data augmentation methods to make it suitable for hyperspectral image classification.

2. We introduce a vision transformer into unsupervised hyperspectral image classification. The transformer model does not have any convolution or recurrent units. The 3D CNNs for hyperspectral image classification need much more computational resources than 2D CNNs for computer vision, and fail to process sequential data well.

We adjust the transformer model parameters to ensure the transformer models are suitable for hyperspectral image processing. Additionally, the transformer used for common computer vision contains at least 12 layers. We apply a two-layer transformer model in this paper, which can reduce the model size for better computational efficiency. The 12-layer transformer for computer vision has 86 million parameters, while our model has much fewer parameters.

3. We combine a contrastive learning method and a transformer model as the framework of our unsupervised model. Our proposed model performs better than traditional representative methods in hyperspectral image classification.

We organize the remainder of the paper as follows: A brief overview of the contrastive learning and transformer model is presented in Section 2. Our proposed model is introduced in Section 3. In Section 4, we provide experimental descriptions and result analysis. Finally, the conclusions are described in Section 5.

## 2. Related Work

### 2.1. Contrastive Learning

Contrastive learning has achieved unprecedented performance in computer vision tasks. This type of unsupervised approach differs from that of representative learning, which executes based on the augmented views of the sample, thus avoiding the computational cost of generating fake samples. Several contrastive learning methods have been proposed, such as similar contrastive learning (SimCLR) [20], momentum contrast for unsupervised visual representation learning (MoCo) [21], and bootstrap your own latent (BYOL) [22].

Among these three methods, both SimCLR and MoCo not only depend on positive pairs but also negative pairs, while obtaining the negative pairs is often a time-consuming process. BYOL, on the contrary, does not need negative pairs; thus, to date, it has achieved the best training efficiency.

The BYOL architecture is shown in Figure 1. BYOL aims to minimize the similarity loss between q(z) and sg(z'). It uses two neural networks: the online and target networks. These two networks have the same architecture: an encoder, a projector, and a predictor, but different weights. After training, only f remains and it can be used for downstream tasks. A detailed introduction of BYOL is presented in Section 3.

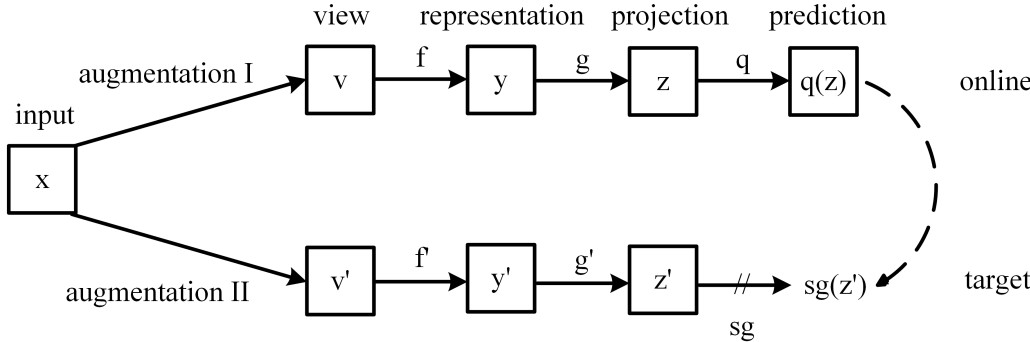

**Figure 1.** The architecture of the BYOL method.

### 2.2. Transformer

The transformer model was originally proposed for natural language processing (NLP) tasks in 2017 [23], and has recently been used for image classification , named "vision transformer" [24]. The transformer model we used in the experiments is shown in Figure 2. We did not make any changes to the transformer model. The transformer model consists of several identical layers. Each layer is composed of two sub-layers, namely, a multi-headed self-attentive mechanism and a fully connected feed-forward network. Each sub-layer has a residual connection followed by layer normalization. Thus, the final output of each sub-

layer can be formulated as *LayerNorm(x + SubLayer(x))*, where *SubLayer(x)* is the function of the sub-layer. The multi-head self-attention is defined as:

$$\text{MultiHead } (Q, K, V) = \text{concat}(\text{ head }_1, \cdots, \text{ head }_h)W^O,\tag{1}$$

where head $_i$ = Attention $\left(QW_i^Q, KW_i^K, VW_i^V\right)$, $W_i^Q \in R^{d_{\text{model}} \times d_q}$, $W_i^K \in R^{d_{\text{model}} \times d_k}$, $W_i^V \in R^{d_{\text{model}} \times d_v}$, and $W^O \in R^{h \times d_v \times d_{\text{model}}}$ are parameter matrices. The attention here is formulated as:

$$\text{Attention } (Q, K, V) = \text{softmax}\left(\frac{QK^T}{\sqrt{d_k}}\right)V,\tag{2}$$

where $Q$ and $K$ of dimension $d_k$ and $V$ of dimension $d_v$ are three learnable weight matrices.

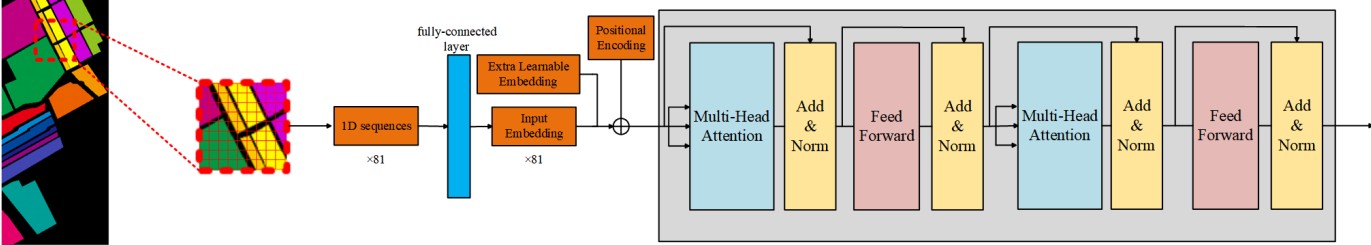

**Figure 2.** The structure of the transformer model used in the experiments. We used a two-layer transformer. The input was $27 \times 27 \times band$. Then, the input image was split into $3 \times 3 \times band$ patches and reshaped into 1D sequences. Next, we obtained the embedding of each sequence via a fully connected layer. Finally, we fed the embeddings to the transformer model after adding the position embeddings to the patch embeddings and an extra learnable embedding.

## 3. Proposed Method

Our proposed framework, as shown in Figure 3, consists of two important parts: a contrastive learning method and a transformer model. These two parts have recently achieved satisfactory results in general image classification. However, these methods cannot be used directly to process hyperspectral images because the spectral information in hyperspectral images cannot be disrupted. Based on the characteristics of the hyperspectral image, we modified the augmentation methods in contrastive learning to make it applicable to hyperspectral images.

Here, we chose BYOL as the contrastive learning method. BYOL uses two neural networks, online and target networks, which learn by interacting with each other. Both neural networks consist of three parts: an encoder f, a projector g, and a predictor q. The online and target networks have the same structure but use different weights. sg is the stop-gradient. First, the online network outputs a representation y, a projection z, and a prediction q(z) from the weakly augmented view of the hyperspectral image, and the target network outputs y' and the target projection z' from the strongly augmented view of the hyperspectral image. Second, the loss between the L2-normalized predictions $\overline{q}(z)$ and target projections $\overline{z'}$ is calculated:

$$L_{\theta,\xi} \triangleq ||\overline{q}(z) - \overline{z'}||_2^2 = 2 - 2 \cdot \frac{\langle q(z), z' \rangle}{||q(z)||_2 \cdot ||z'||_2},\tag{3}$$

where $\langle \cdot, \cdot \rangle$ is the inner product. To symmetrize the loss $L_{\theta,\xi}$, $L'_{\theta,\xi}$ is computed by feeding the strongly augmented view of the hyperspectral image into the online network and the weakly augmented view of the hyperspectral image into the target network. The final loss is formulated as $L_{\theta,\xi}^{\text{BYOL}} = L_{\theta,\xi} + L'_{\theta,\xi}$. At each training step, BYOL minimizes the loss with respect to $\theta$ only, but $\xi$ is a slowly moving exponential average of $\theta$: $\xi \leftarrow \tau\xi + (1 - \tau)\theta$, where $\tau$ is a target decay rate.

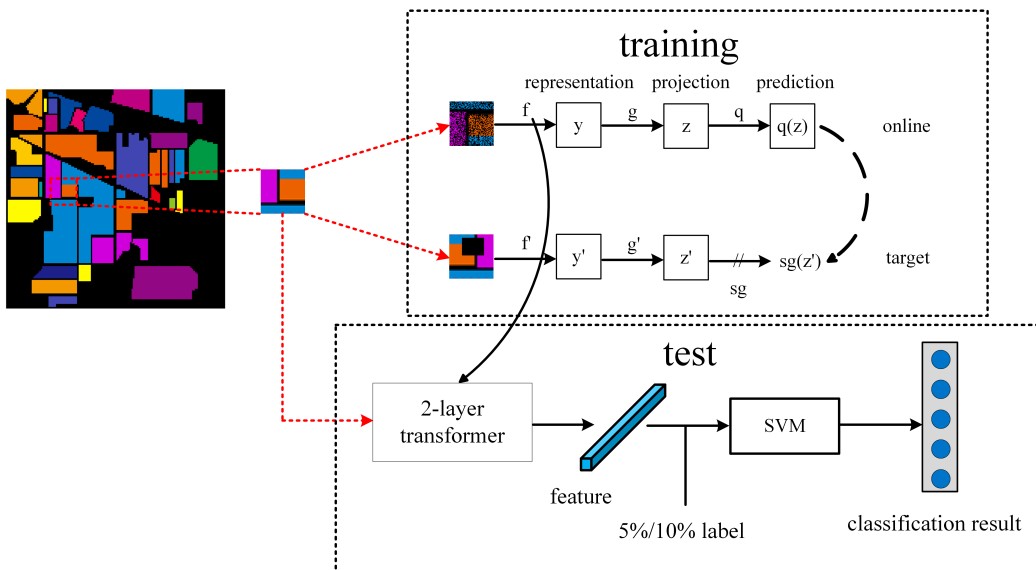

**Figure 3.** The overall architecture of our proposed framework.

It was empirically shown that the combination of adding the predictor to the online network and using the moving average of the online network parameters as the target network encourages encoding more and more information within the online projection and avoids collapsed solutions, such as constant representations. In this study, we reduced the model size for computational efficiency. Additionally, we applied a new image augmentation method for hyperspectral images.

The two views of the hyperspectral image used in BYOL are differently augmented images. In our model, we take the horizontal flip or vertical flip as the preliminary augmentation method, and different random erasures after that as the different augmentation method. Random erasure can be divided into two types: random rectangular area erasure and random point erasure; neither of them erase the center point, as shown in Figure 4. The procedure of selecting the rectangle area and erasing this area is presented in Algorithm 1, and the procedure of selecting the points and erasing these points is shown in Algorithm 2.

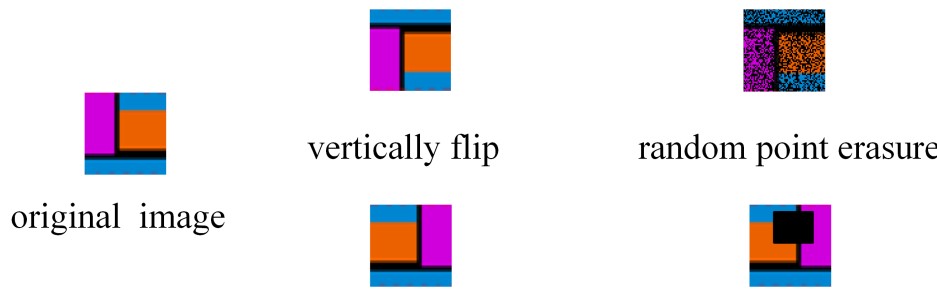

**Figure 4.** The image augmentation methods of our proposed framework.

---

**Algorithm 1** Random rectangular area erasure procedure

---

**Input:** Input image $I$;
**Output:** Erased image $I^*$.
**1** Generate a matrix the same size as the input image $I$ using 1.
**2** Select a random sub-matrix in this matrix and change the elements inside to 0.
**3** If the center point of the matrix is in the sub-matrix, change the element of that point to 1.
**4** Multiply the image by this matrix to obtain the erased image $I^*$.
**5** Return the erased image $I^*$.

---

---

**Algorithm 2** Random point erasure procedure

---

**Input:** Input image $I$;
**Output:** Erased image $I^*$.
**1** Generate a random matrix the same size as the image $I$ using 0 and 1 with the same probability.
**2** If the center point of the matrix is 0, change the element of that point to 1.
**3** Multiply the image by this matrix to obtain the erased image $I^*$.
**4** Return the erased image $I^*$.

---

Furthermore, we employed the transformer model as the encoder instead of a CNN. In a transformer model, the input image is split into fixed-size patches and then reshaped into 1D sequences. Next, a fully connected layer is used to obtain the embedding of each sequence. After adding the position embedding to these sequences and an extra learnable sequence, the sequences are fed to a two-layer transformer encoder. After getting the output sequences, the first one is used for further process. Unlike the vision transformer, we remove the MLP head and apply a small patch size. Additionally, our proposed model consists many fewer layers than the vision transformer, which consists of at least 12 layers. As shown in Figure 2, to reduce the computational resource consumption, we only use a two-layer model. The 12-layer transformer for computer vision has 86 million parameters, whereas our model has many fewer parameters, as shown in Table 1.

After training, we can use the two-layer transformer model to obtain the features of an HSI image under the unsupervised condition. Then, based on the features, with a small proportion of the label information, a simple SVM can achieve high accuracy.

**Table 1.** The parameters of our transformer.

| Layers | Dim | Heads | MLP Dim | Head Dim | Parameters |
|:---:|:---:|:---:|:---:|:---:|:---:|
| 2 | 256 | 16 | 512 | 32 | 1,612,288 (SV&UP) 1,646,848 (IP) |

## 4. Experimental Descriptions and Result Analysis

In this section, the experimental descriptions and result analysis are presented.

### 4.1. Datasets' Description

The experiments were conducted on three publicly available datasets, Indian Pines (IP), University of Pavia (UP), and Salinas Scene (SV), as indicated in Figure 5.

The IP dataset was collected by the AVIRIS sensor in Northwestern Indiana. The scene size is $145 \times 145$. After removing the bands absorbed by water, the number of bands reduces to 200. The 10,249 labeled pixels are designed into 16 classes.

The UP scene was gathered by the ROSIS sensor over Pavia, Northern Italy. UP has $610 \times 610$ pixels. Each pixel consists of 103 spectral bands. The 42,776 labeled pixels are divided into nine categories.

The SV image was acquired by the AVIRIS sensor over Salinas Valley, California. The image size is $512 \times 217$ with 204 available spectral bands. The 54,129 labeled pixels are partitioned into 16 categories.

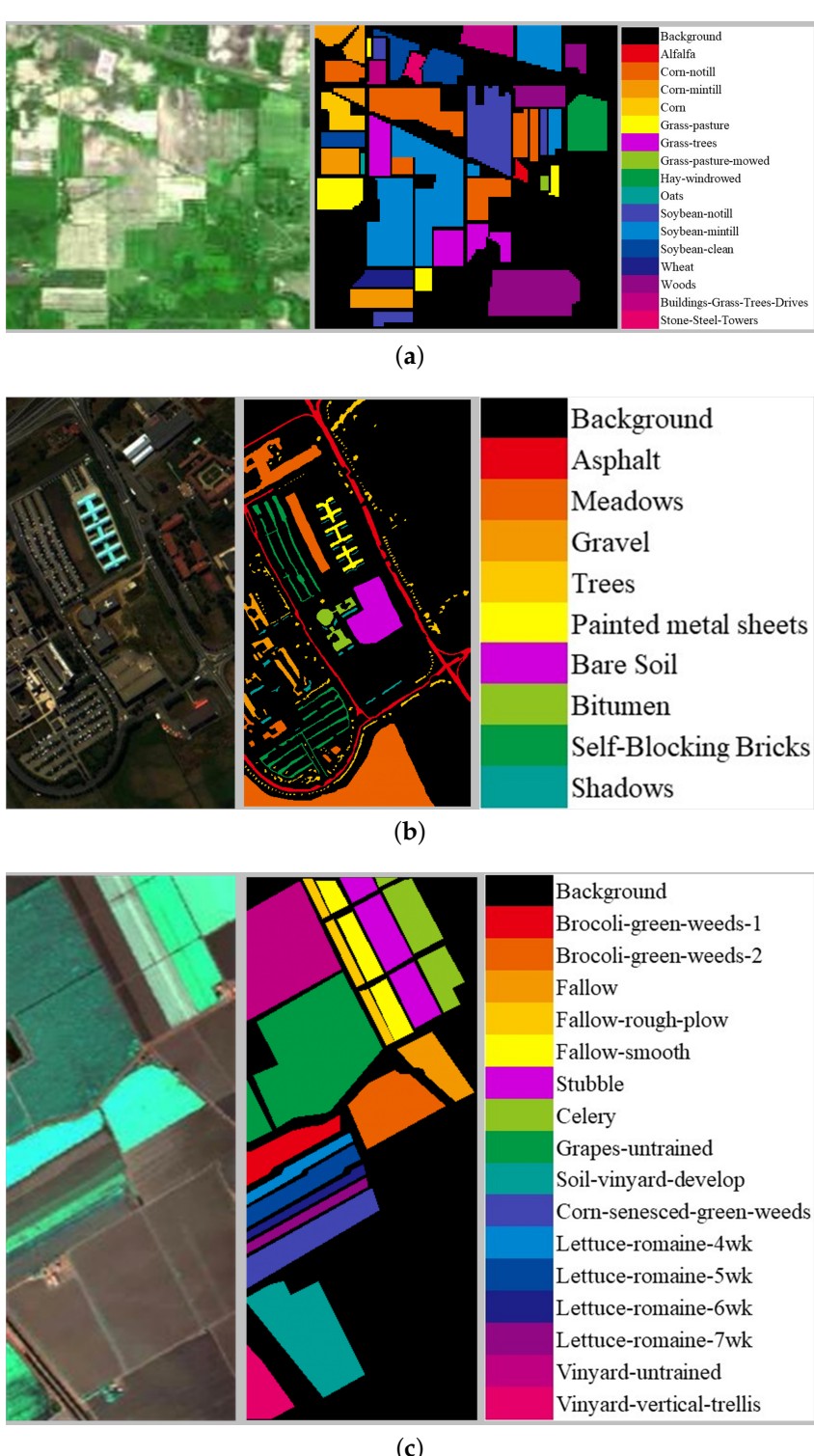

**Figure 5.** (**a**–**c**) The false-color image and ground-truth map of the IP, UP, and SV datasets.

*4.2. Experimental Parameters*

The parameters of our model are shown in Tables 1 and 2.

**Table 2.** The structures of our projector and predictor.

| | Projector | | | Predictor | |
|---|---|---|---|---|---|
| Layer (Type) | Output Shape | Parameters | Layer (Type) | Output Shape | Parameters |
| Linear | [−1, 1024] | 263,168 | Linear | [−1, 1024] | 66,560 |
| BatchNorm1d | [−1, 1024] | 2048 | BatchNorm1d | [−1, 1024] | 2048 |
| ReLU | [−1, 1024] | 0 | ReLU | [−1, 1024] | 0 |
| Linear | [−1, 64] | 65,600 | Linear | [−1, 64] | 65,600 |

All experiments were performed on a Titan-RTX GPU. The model was implemented in Python using the Pytorch framework. We adopted a traditional principal component analysis (PCA) to remove the spectral redundancy. The input size is $27 \times 27 \times 30$ for IP and $27 \times 27 \times 15$ for UP and SV. We set the patch size to 3 and the batch size to 256. The target decay rate was 0.99. The learning rate was set to 0.003. We trained the transformer model for 20 epochs, and chose the model with the least loss for the test.

For IP and UP, the proportion of samples for training was set to 10%. For SV, we selected 5% of the samples for training.

### 4.3. Result Analysis

In this study, we used the overall accuracy (OA) and average accuracy (AA) as performance evaluation metrics. OA is the proportion of correctly classified samples to all samples; AA represents the average of classification accuracy for each category. We adopted six other methods as baselines. These models contain three supervised methods: linear discriminant analysis (LDA) [25], deep convolutional neural network (1D-CNN) [26], and supervised deep feature extraction (S-CNN) [27]; and three unsupervised methods: 3D convolutional autoencoder(3DCAE) [16], adversarial autoencoder [28], and variational autoencoder [28].

The classification results with IP are shown in Table 3 and Figure 6. These results show that our model achieves the best performance with IP. As IP is more difficult to be classified among these three public datasets, we conclude that our model is superior to the others. The classification results with SV are presented in Table 4 and Figure 7. Similar to IP, our proposed method performed the best in 10 classes and approached the best performance in the other six classes. However, according to the results in UP demonstrated in Table 5 and Figure 8, the AA of our model is second only to the AA of AAE among all seven models. We presume this is due to the small number of training samples in the last category. Considering the overall performance, our findings demonstrate the feasibility of deep-learning-based hyperspectral image classification without convolution. Based on the above experimental results, the transformer is a promising model for hyperspectral image classification that warrants further investigation.

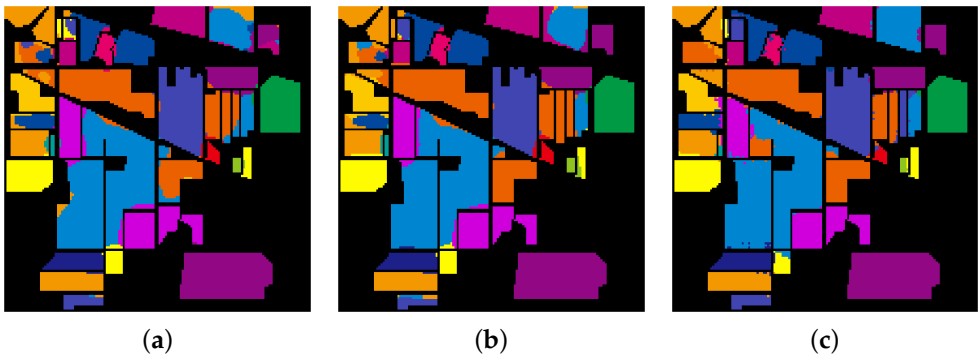

(a)    (b)    (c)

**Figure 6.** The classification maps of IP. (**a**–**c**) Classification results of AAE, VAE, and our proposed model, respectively.

**Table 3.** Classification results of different models in IP. Some of the data in the table were obtained from [16]. The best values in each row are bolded.

| Class | Supervised Feature Extraction | | | Unsupervised Feature Extraction | | | |
|---|---|---|---|---|---|---|---|
| | LDA | 1D-CNN | S-CNN | 3DCAE | AAE | VAE | Proposed |
| 1 | 58.54 | 43.33 | 83.33 | 90.48 | **100.00** | **100.00** | 97.05 |
| 2 | 69.88 | 73.13 | 81.41 | 92.49 | 81.63 | 78.78 | **96.73** |
| 3 | 65.86 | 65.52 | 74.02 | 90.37 | 95.27 | 92.37 | **95.34** |
| 4 | 73.71 | 51.31 | 71.49 | 86.90 | **99.22** | 97.34 | 98.97 |
| 5 | 90.32 | 87.70 | 90.11 | 94.25 | **95.17** | 93.87 | 94.82 |
| 6 | 92.09 | 95.10 | 94.06 | 97.07 | **98.73** | 98.27 | 93.85 |
| 7 | 96.00 | 56.92 | 84.61 | 91.26 | 96.00 | **98.67** | 95.00 |
| 8 | 98.14 | 96.64 | 98.37 | 97.79 | **99.84** | 99.77 | 98.85 |
| 9 | 11.11 | 28.89 | 33.33 | 75.91 | 96.30 | 98.15 | 88.88 |
| 10 | 73.80 | 75.12 | 86.05 | 87.34 | 87.01 | 78.86 | **95.65** |
| 11 | 55.41 | 83.49 | 82.98 | 90.24 | 89.08 | 81.75 | **98.56** |
| 12 | 76.92 | 67.55 | 73.40 | **95.76** | 93.51 | 90.64 | 94.31 |
| 13 | 91.30 | 96.86 | 87.02 | 97.49 | 98.56 | 98.56 | 89.37 |
| 14 | 93.32 | 96.51 | 94.38 | 96.03 | 95.73 | 93.24 | **98.44** |
| 15 | 67.72 | 39.08 | 75.57 | 90.48 | 97.31 | 97.02 | **99.70** |
| 16 | 90.36 | 89.40 | 79.76 | 98.82 | 98.02 | **98.81** | 94.73 |
| AA(%) | 76.89 | 71.66 | 84.44 | 92.04 | 95.09 | 93.51 | **95.64** |
| OA(%) | 76.88 | 79.66 | 80.72 | 92.35 | 91.80 | 88.03 | **96.78** |

**Table 4.** Classification results of different models in SV. Some of the data in the table were obtained from [16]. The best values in each row are bolded.

| Class | Supervised Feature Extraction | | | Unsupervised Feature Extraction | | | |
|---|---|---|---|---|---|---|---|
| | LDA | 1D-CNN | S-CNN | 3DCAE | AAE | VAE | Proposed |
| 1 | 99.16 | 97.98 | 99.55 | **100.00** | 99.98 | 99.74 | **100.00** |
| 2 | 99.94 | 99.25 | 99.43 | 99.29 | **100.00** | 99.79 | **100.00** |
| 3 | 99.79 | 94.43 | 98.81 | 97.13 | **100.00** | **100.00** | **100.00** |
| 4 | **99.77** | 99.42 | 97.45 | 97.91 | 99.09 | 99.57 | 98.17 |
| 5 | 98.98 | 96.60 | 97.96 | 98.26 | 99.42 | **99.71** | 98.55 |
| 6 | 99.89 | 99.51 | 99.83 | **99.98** | 99.97 | 99.86 | 99.49 |
| 7 | 99.97 | 99.27 | 99.59 | 99.64 | 99.93 | 99.96 | **100.00** |
| 8 | 81.84 | 86.79 | 94.40 | 91.58 | 91.21 | 86.91 | **100.00** |
| 9 | 99.90 | 99.08 | 98.85 | 99.28 | 99.69 | **99.99** | 99.83 |
| 10 | 96.31 | 93.71 | 97.35 | 96.65 | 98.46 | 96.54 | **100.00** |
| 11 | 99.61 | 94.55 | 97.71 | 97.74 | 99.57 | **100.00** | 98.92 |
| 12 | 99.67 | 99.59 | 98.73 | 98.84 | **100.00** | 99.44 | **100.00** |
| 13 | 99.20 | 97.50 | 96.72 | 99.26 | **99.89** | 97.24 | 98.97 |
| 14 | 96.56 | 94.08 | 95.22 | 97.49 | 99.08 | 96.49 | **100.00** |
| 15 | 73.60 | 66.52 | 95.61 | 87.85 | 93.69 | 87.90 | **99.43** |
| 16 | 98.48 | 97.48 | 99.44 | 98.34 | 99.73 | 99.61 | **100.00** |
| AA(%) | 96.42 | 94.73 | 97.39 | 97.45 | 98.73 | 97.67 | **99.58** |
| OA(%) | 92.18 | 91.30 | 97.92 | 95.81 | 97.10 | 95.23 | **99.71** |

**Table 5.** Classification results of different models in UP. Some of the data in the table were obtained from [16]. The best values in each row are bolded.

| Class | Supervised Feature Extraction | | | Unsupervised Feature Extraction | | | |
|---|---|---|---|---|---|---|---|
| | LDA | 1D-CNN | S-CNN | 3DCAE | AAE | VAE | Proposed |
| 1 | 78.41 | 90.93 | 95.40 | 95.21 | 95.39 | 81.31 | **98.19** |
| 2 | 83.69 | 96.94 | 97.31 | 96.06 | 98.96 | 97.65 | **99.86** |
| 3 | 73.02 | 69.43 | 81.21 | 91.32 | 97.49 | 91.00 | **98.87** |
| 4 | 93.68 | 90.32 | 95.83 | 98.28 | 96.94 | 94.79 | **96.99** |
| 5 | **100.00** | 99.44 | 99.91 | 95.55 | 99.94 | 99.94 | 96.75 |
| 6 | 88.51 | 73.69 | 95.29 | 95.30 | 99.29 | 99.91 | **100** |
| 7 | 85.75 | 83.42 | 87.05 | 95.14 | **100.00** | 99.42 | 96.32 |
| 8 | 74.49 | 83.65 | 87.35 | 91.38 | 97.90 | 94.77 | **98.32** |
| 9 | 99.11 | 98.23 | 95.66 | **99.96** | 96.32 | 88.62 | 84.04 |
| AA(%) | 86.29 | 87.34 | 94.75 | 95.36 | **98.03** | 94.16 | 96.59 |
| OA(%) | 83.75 | 89.99 | 92.78 | 95.39 | 98.14 | 94.53 | **98.67** |

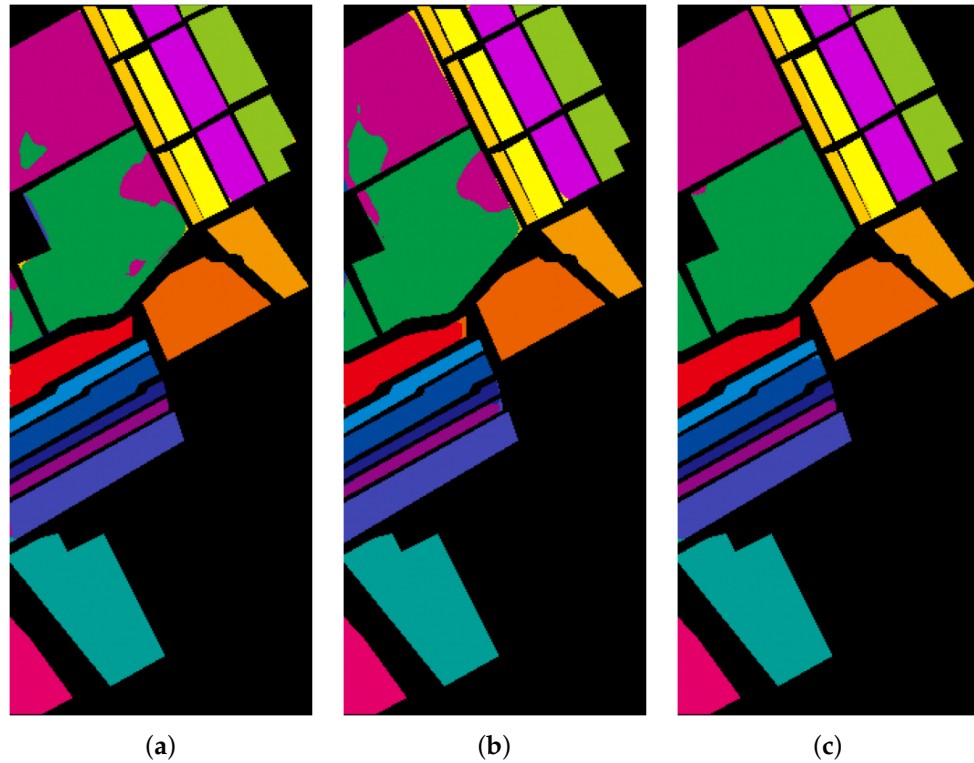

(a)            (b)            (c)

**Figure 7.** The classification maps of SV. (**a**–**c**) Classification results of AAE, VAE, and our proposed model, respectively.

Even though our model works without label information, it still outperformed the supervised methods. Additionally, compared to the LDA results, we conclude that the deep-learning-based model is superior to the machine-learning-based method.

According to the above experimental results and analysis, we conclude that our model can effectively extract the features under unsupervised conditions. It is suitable for addressing the challenge posed by the lack of labeling. The model does not contain convolutional operations. Our findings prove that convolutional operations are not necessary for hyperspectral image classification. Additionally, 2D convolutional networks for traditional computer vision are not applicable to hyperspectral image classification. A 3D convolutional network can provide a better hyperspectral image classification result than a 2D convolutional network. The transformer model we used in the experiments is the same as the transformer model used for traditional computer vision tasks in terms of model

structure. We only changed the model size, which demonstrates that computer vision models have huge potential for hyperspectral image classification. The other important part of our method, contrastive learning, relies on data augmentation for the input image. Because some data argumentation methods in traditional computer vision are unsuitable for hyperspectral images, we used image flip, and deleting spectral information of some points to augment the data. With a small number of data augmentation methods, the contrastive learning method can perform much better than the representative learning method. With more data augmentation methods, the contrastive learning method may be more accurate for hyperspectral image classification.

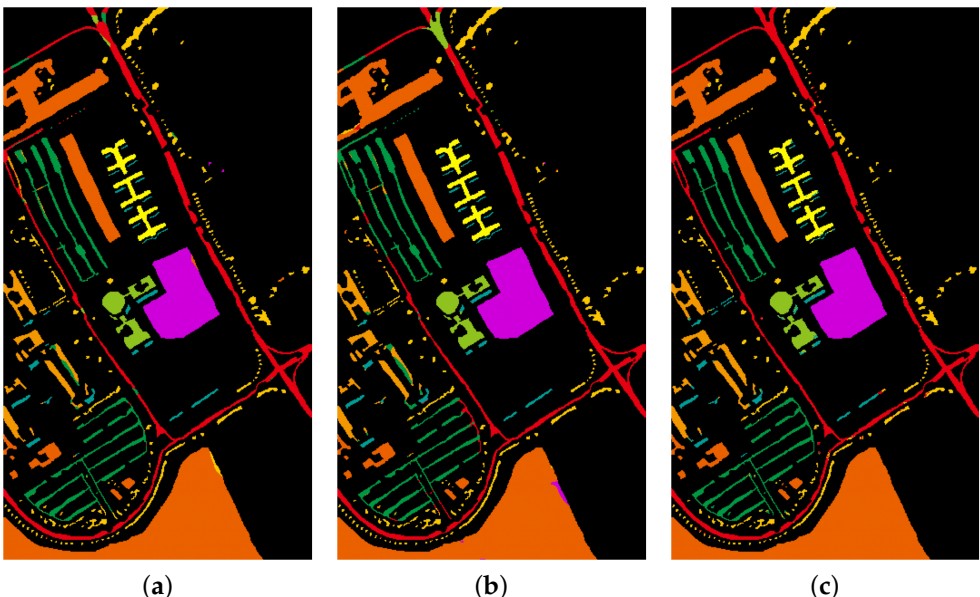

(a)　　　　　　　　　　(b)　　　　　　　　　　(c)

**Figure 8.** The classification maps of UP. (**a**–**c**) Classification results of AAE, VAE, and our proposed model, respectively.

## 5. Conclusions

In this paper, we proposed an unsupervised framework based on a transformer and contrastive learning. Both methods can be used to reduce computational resource consumption. The experiments with three publicly available data sets demonstrated the more accurate performance of the proposed method compared to other methods. As the transformer for the visual tasks and contrastive learning method are widely used in computer vision, we think our proposed method has great potential for hyperspectral image processing.

**Author Contributions:** X.H. and T.Z. implemented the algorithms, designed the experiments, and wrote the paper; Y.L. performed the experiments; Y.P. and T.L. guided the research. All authors have read and agreed to the published version of the manuscript.

**Funding:** This research was partially supported by the National Key Research and Development Program of China (Nos. 2017YFB1301104 and 2017YFB1001900), the National Natural Science Foundation of China (Nos. 91648204 and 61803375), and the National Science and Technology Major Project.

**Institutional Review Board Statement:** Not applicable.

**Informed Consent Statement:** Not applicable.

**Data Availability Statement:** The datasets involved in this paper are all public datasets. They can be downloaded at http://www.ehu.eus/ccwintco/index.php/Hyperspectral_Remote_Sensing_Scenes accessed on 5 August 2021.

**Acknowledgments:** The authors acknowledge the State Key Laboratory of High-Performance Computing, College of Computer, National University of Defense Technology, China.

**Conflicts of Interest:** The authors declare no conflict of interest.

## Abbreviations

The following abbreviations are used in this manuscript:

| | |
|---|---|
| HSI | Hyperspectral Image |
| CNN | Convolutional Neural Network |
| GAN | Generative Adversarial Network |
| NLP | Natural Language Processing |
| SimCLR | Similar Contrastive Learning |
| MoCo | Momentum Contrast for Unsupervised Visual Representation Learning |
| BYOL | Bootstrap Your Own Latent |

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
