# Peer review of "Contrastive Learning Based on Transformer for Hyperspectral Image Classification"

_applsci, doi:10.3390/app11188670_

Round 1

Reviewer 1 Report

Introduction provide sufficient background and are include all relevant references. Paper is organised in appropriated design. The used methods are adequately described. The results are clearly presented. The Conclusions are clearly presented and are supported by the results.

Reviewer 2 Report

The authors proposed a method for hyperspectral image classification. The article is interesting but needs some improvements as mentioned below:

  • The method BOYL is not explained in the related work section.
  • Overall architecture in related work section doesn't make sense.
  • Architecture diagrams need improvement for better understanding and clarity.
  • I cannot understand, how vertical flip is applied in figure 3.
  • In proposed method, how model size is reduced for computational efficiency?
  • The article should be proofread for any grammatical and spell errors. 
